# Origin of circulating free DNA in patients with lung cancer

**Tomonori Abe**[1], **Chiho Nakashima**[1], **Akemi Sato**[2], **Yohei Harada**[1], **Eisaburo Sueoka**[2], **Shinya Kimura**[1], **Atsushi Kawaguchi**[3], **Naoko Sueoka-Aragane**[1]*

1 Division of Hematology, Respiratory Medicine and Oncology, Department of Internal Medicine, Faculty of Medicine, Saga University, Saga, Japan, 2 Department of Clinical Laboratory Medicine, Faculty of Medicine, Saga University, Saga, Japan, 3 Education and Research Center for Community Medicine, Faculty of Medicine, Saga University, Saga, Japan

* sueokan@cc.saga-u.ac.jp

## Abstract

Liquid biopsy has become widely applied in clinical medicine along with the progress in innovative technologies, such as next generation sequencing, but the origin of circulating tumor DNA (ctDNA) has not yet been precisely established. We reported bimodal peaks of long fragment circulating free DNA (cfDNA) of 5 kb and short fragment cfDNA of 170 bp in patients with advanced lung cancer, and both contained ctDNA. In this paper, we demonstrate that the total amount of cfDNA is higher when patients with lung cancer have extra-thoracic metastases, and the amount of long fragment cfDNA is significantly higher in those patients. To investigate the origin of long fragment cfDNA, conditioned media isolated from lung cancer cell lines was fractionated. Long fragment cfDNA was found concomitant with extracellular vesicles (EVs), but short fragment cfDNA was not observed in any fractions. However, in peripheral blood from a metastatic animal model both fragments were detected even with those same lung cancer cell lines. In human plasma samples, long fragment cfDNA was observed in the same fraction as that from conditioned media, and short fragment cfDNA existed in the supernatant after centrifugation at 100,000g. Concentration of ctDNA in the supernatant was two times higher than that in plasma isolated by the conventional procedure. Long fragment cfDNA associated with tumor progression might therefore be released into peripheral blood, and it is possible that the long fragment cfDNA escapes degradation by co-existing with EVs. Examination of the biological characteristics of long fragment cfDNA is a logical subject of further investigation.

## Introduction

Liquid biopsy using circulating tumor DNA (ctDNA) isolated from peripheral blood has been clinically applied in cancer treatment, including molecular targeted therapy, and its use has spread worldwide [1]. In patients with non-small cell lung cancer (NSCLC) bearing epithelial growth factor receptor (*EGFR*) activating/sensitive mutations, such as L858R and exon 19 deletions, *EGFR* tyrosine kinase inhibitors (EGFR-TKIs) have been highly effective in

**Data Availability Statement:** All relevant data are within the paper and its Supporting Information files.

**Funding:** NA received funding support. This work was supported in part by Grants-in Aid for Cancer

Research: Special Cancer Research, from the Ministry of Education, Culture, Science, and Technology, Japan (https://www.jsps.go.jp/j-grantsinaid); JP17K07197. The funders had no role in study design, data collection and analysis, decision to publish, or preparation of the manuscript. There was no additional external funding received for this study.

**Competing interests:** The authors have declared that no competing interests exist.

approximately 70% of cases [2,3]. Detection of *EGFR* mutations with ctDNA in NSCLC has been approved as a companion diagnostic for EGFR-TKIs. As technologies of molecular biology have rapidly progressed, sensitivity of mutation detection has improved to 0.03% with the next generation sequencing (NGS) technique [4–6]. However, whether liquid biopsy is useful in early stages of cancer has not been clarified, since the amount of ctDNA is associated with tumor burden. Achieving further progress at detection requires improved isolation system of ctDNA. It is therefore important to examine mechanisms of how ctDNA is released from tumors and how to effectively isolate it.

Circulating free DNA (cfDNA) exists in peripheral blood and is derived from tumor cells and from normal cells (mainly lymphocytes). Recently, we reported that cfDNA was better isolated with cellulose magnetic beads for DNA capturing, and the amount of ctDNA was greater than that obtained with silica membrane spin columns [7]. Using the capillary electrophoresis system, we also demonstrated the existence of bimodal peaks, comprising long fragment cfDNA of 5Kb and short fragment cfDNA of 170bp. We also have shown that poor pre-analytical procedures—such as inadequate plasma storage—caused the appearance of long fragment cfDNA [7,8]. However, long fragment cfDNA was still observed even after appropriate pre-analytical procedures, but only in patients with advanced lung cancer, not in patients with early stage lung cancer or healthy volunteers. Therefore, we assumed that long fragment cfDNA could be related to tumor progression, but its origin remains in doubt. Short fragment cfDNA has been thought to be an apoptotic product considering that it is the same size as the nucleosome unit [9]. Oncosomes at the size of 1–10 μm among EV have been known to appear in peripheral blood of patients with advanced cancer, and primarily contain long fragment cfDNA [10].

The rapid progress of technology for liquid biopsy has led to high sensitivity of ctDNA detection, but it is still limited to advanced cancers. It is now necessary to improve the effectiveness of the ctDNA isolation system to allow detection even among patients who are in the early stages of cancer. Based on our previous results, we hypothesized that long fragment cfDNA observed in patients with advanced lung cancer might be contained in oncosomes, so that purification of ctDNA derived from oncosomes could lead to improved sensitivity for detecting ctDNA. In this paper, we report the results of our investigation into the origin of cfDNA, especially that related with EVs, and discuss how to isolate ctDNA effectively.

## Materials and methods

### Cell lines

Human lung cancer cell lines H1975 (ATCC® CRL-5905™), H838 (ATCC® CRL-5844™), and H226B were purchased from American Type Culture Collection (Manassas, VA). H1975 cells, which carry the *EGFR* L858R/T790M mutation, were cultured in RPMI-1640 containing high glucose, L-glutamine, and HEPES (ATCC #30–2001; American Type Culture Collection, Manassas, VA), supplemented with 10% fetal bovine serum, at 37˚C in 5% carbon dioxide. H838 cells, which carry the *EGFR* L858R mutation, were cultured in RPMI-1640 supplemented with 10% fetal bovine serum at 37˚C in 5% carbon dioxide. H226B cells, which are a human squamous cell lung cancer cell line and do not carry any *EGFR* mutations, were cultured in RPMI-1640 supplemented with 10% fetal bovine serum at 37˚C in 5% carbon dioxide.

### Patient selection

Peripheral blood specimens were obtained from 92 patients with lung cancer, 18 patients with benign pulmonary disease, and 20 healthy individuals. All patients were examined at a Saga University Hospital, from May 2016 to November 2016; volunteers were recruited from

among Saga University Hospital staff. The analysis was retrospective. Pathological stage was determined according to the current 7[th] edition of the TNM classification. The Clinical Research Ethics Committee of Saga University Hospital approved the study protocol. All participants gave informed consent for blood sample collection, and all testing was done according to the Declaration of Helsinki.

## Extraction of DNA

Peripheral blood specimens were collected into tubes containing 3.2% sodium citrate (product number VP-CA052K; TERUMO CORPORATION, Tokyo, Japan). Plasma was immediately separated from blood cells by 3,000 rpm (1,710 g) centrifugation at 4˚C for 20 min. Plasma was collected and stored at −80˚C until assays were performed. DNA was isolated from 1 ml plasma using the Maxwell RSC ccfDNA plasma cartridge[®] (Product no. AS1480; Promega, Mannheim, Germany) according to the manufacturer's instructions. Each DNA sample extracted by Maxwell RSC was eluted with 60 μl TE buffer. All DNA samples were stored at −20˚C until further examination.

## Quantification of DNA

DNA concentration was quantified with the QuantiFluor[®] dsDNA system and a Quantus[®] Fluorometer (Promega, Mannheim, Germany). The QuantiFluor[®] dsDNA System provides a fluorescent DNA-binding dye that enables sensitive and specific quantitation of small amounts of double-stranded DNA (dsDNA) in solution. The dye shows minimal binding to single-stranded DNA (ssDNA) and RNA, allowing specific quantitation of dsDNA. All experiments were performed three times.

## Size distribution analysis of DNA

The size distribution of DNA was examined with a capillary electrophoresis system. We used the High Sensitivity DNA kit (Agilent Technologies Inc., Santa Clara, CA, Product no. 5067–4626), a microchip, and analyzed it with an Agilent 2100 Bioanalyzer[®] equipped with Expert 2100 software (Agilent Technologies Inc., Santa Clara, CA) according to the manufacturer's instructions. We defined "short fragment cfDNA" as DNA of size 120~265 bp and "long fragment cfDNA" as 1000~9000 bp, as reported previously [7]. The concentration and molarity of each specified region were automatically calculated against internal standards by Agilent 2100 Expert software version B.02.08.

## Isolation of EVs from conditioned media

Conditioned media from cancer cell lines was collected after 24 h incubation in media with exosome-depleted FBS (System Biosciences, Palo Alto, CA, USA). Cells were eliminated by centrifugation at 3,000 rpm (1,710 g) for 20 min. Then the supernatant was centrifuged at 10,000 g for 30 min and we obtained the materials (10K pellet). After that, the supernatant was centrifuged at 100,000 g for 60 min and we obtained those pelleted materials (100K pellet). For this step, we used an Optima L-70 ultracentrifuge (Beckman Coulter Inc, USA) with an SW-40i rotor.

## Transmission electron microscope (TEM)

**Negative staining.** The samples were absorbed to formvar film coated copper grids and stained with 2% phosphotungstic acid solution (pH 7.0) for 60 sec.

**Observation and imaging.** The grids were observed with a transmission electron microscope (JEM-1400Plus; JEOL Ltd., Tokyo, Japan) at an acceleration voltage of 100 kV. Digital images (3296×2472 pixels) were taken with a CCD camera (EM-14830RUBY2; JEOL Ltd., Tokyo, Japan).

## Western blotting

The fraction containing EVs was washed with PBS, and lysed with buffer containing Tris-HCl (pH 8.0), 150 mM NaCl, 1% Triton X-100, 0.1% sodium dodecyl sulfate, and 1% sodium deoxycholate with both phosphatase and protease inhibitors. Western blotting was conducted by use of the NuPAGE electrophoresis system (NOVEX, San Diego, CA) according to the manufacturer's instructions. The proteins were fractionated by 10% Bis-Tris gel electrophoresis followed by transfer to nitrocellulose membrane (Schleicher & Schuell, Inc., Keene, NH). After blocking with 5% skim milk, we used the following antibodies for western blotting: mouse monoclonal anti-CD9 antibody (1:1000 dilution; COSMO BIO, Tokyo, Japan), mouse monoclonal anti-CD63 antibody (1:500 dilution; COSMO BIO, Tokyo, Japan), BiP antibody (1:1000 dilution; Cell Signaling #3177), anti Annexin A1 antibody (1:2000 dilution; abcam, ab214486). The ImmunoStar$^{®}$ LD or ImmunoStar$^{®}$ Zeta was used for detection (FUJIFILM Wako Pure Chemical Corporation, Osaka, Japan).

## Animal experiment

NOD/SCID/JAK3$^{null}$ (NOJ) mice were established as described previously [11]. They were housed under pathogen-free conditions in animal facilities at Saga University according to institutional guidelines. The protocol was approved by the Committee on the Ethics of Animal Experiments of the Saga University (Permit Number: 24-037-0) [12]. All surgery was performed under 2% of isoflurane anesthesia, and all possible efforts were made to minimize suffering. All research staff were provided with training in animal care or handling.

H1975 cells ($1×10^7$) were injected into the dorsal flanks of 24 NOJ mice. Tumor volumes, defined as [(short axis) $^2$ × (long axis)]/2, and body weight were measured every 3 days. Mice were evaluated daily for signs of morbidity or tumor growth. When body weight decreased by 10% of original weight, or metastatic manifestations—such as superficial lymph node swelling or ascites—were observed, the animal was euthanized by cervical dislocation and dissected within 24 h. No mice died before meeting criteria for euthanasia. Six mice were sacrificed monthly from one to four months after injection to evaluate metastatic status and size distribution of cfDNA. When we dissected the mice, we examined the lungs, liver, brain, lymph nodes, ascites, and pleural effusion for metastases. Peripheral blood samples from the mice were collected into tubes containing 3.2% sodium citrate. Plasma was immediately separated from blood cells by 3,000 rpm (1,710 g) centrifugation at 4˚C for 20 min. Supernatants were collected and stored at −80˚C until assays could be performed. DNA was isolated from 200 µl of plasma with a QIAamp DNA mini kit (QIAGEN, Hilden, Germany) according to the manufacturer's instruction. The DNA concentration was quantified with the QuantiFluor$^{®}$ dsDNA system and a Quantus$^{®}$ Fluorometer. The size distribution of DNA was examined with a capillary electrophoresis system.

## Separation of plasma DNA into short and long fragment cfDNA by centrifugation

One ml of plasma was centrifuged by 10,000 g at 4˚C for 30 min and we obtained the pelleted materials (10K pellet) as large EVs. Then the supernatant was centrifuged at 100,000g for 60 min and we obtained those pelleted materials (100K pellet) as small EVs (exosomes) and the

supernatant as cell-free plasma. DNA was isolated with a Maxwell RSC ccfDNA plasma cartridge® according to the manufacturer's instructions. The size distribution of DNA was examined with a capillary electrophoresis system.

### Detection of *EGFR* L858R mutation

*EGFR* L858R mutation was detected with a QX200 Droplet Disital PCR (ddPCR) system (Bio-Rad, Hercules, CA, USA). Reaction mixtures of L858R mutation for ddPCR were assembled from supermix for probes (No dUTP) (Bio-Rad, Hercules, CA, USA), PrimePCR ddPCR Mutation Assay:*EGFR* p.L858R, Human (Bio-Rad, Hercules, CA, USA), PrimePCR ddPCR Mutation Assay:*EGFR* WT for p.L858R,Human (Bio-Rad, Hercules, CA, USA), and template DNA. Reaction mixtures were loaded into sample wells and analysis was performed with the Quantasoft Software (Bio-Rad, Hercules, CA, USA) according to the manufacturer's instructions.

### Statistical analysis

Statistical analysis was conducted with SPSS version 19 (IBM SPSS Statistics, IBM, Tokyo, Japan) and JMP® analysis (SAS Institute Inc., NC, US). Comparisons among three or four groups were made with the nonparametric Kruskal-Wallis test; $P < 0.05$ was considered statistically significant. Comparisons between two groups were made with the nonparametric Mann-Whitney U test; $P < 0.05$ was considered statistically significant. If the outcome was statistically significant, multiple pairwise comparisons were made with the Steel-Dwass test. The Hodges-Lehmann method was used to estimate median differences and 95% confidence intervals (CIs) for compariing the two groups. Correlation between two groups was assessed with the non-parametric Spearman correlation coefficient.

## Results

### Total cfDNA higher in lung cancer with extrathoracic metastases than without

Study participants are summarized in Table 1. Median cfDNA concentration in 92 patients with lung cancer, 18 patients with benign pulmonary diseases, and 20 healthy individuals was 10.1, 8.0, and 7.5 ng/ml plasma, respectively (Fig 1A), and the difference between lung cancer and the others was not statistically significant. It was somewhat higher in pathological stage IV, but again the difference was not statistically significant (Fig 1B). Since we have reported that the frequency of ctDNA was higher in distant metastasis than in the intrathoracic region on the basis of a prospective observational study [13], we compared patients with and without extrathoracic metastases. Among patients with lung cancer the median concentration of total cfDNA in those with extrathoracic metastases was significantly higher than in those without metastasis (15.0 ng/ml plasma vs. 9.1 ng/ml plasma), Hodges-Lehmann estimate 6.1 (95%CI 1.5–11.1) p<0.01 (Fig 1C). Since we have recently shown that the size distribution of cfDNA fragments in patients with lung cancer is bimodal with peaks around 170 bp (short) and 5 Kb (long) [7], we next analyzed with capillary electrophoresis the amount of cfDNA fragments in each size class among the patients with lung cancer. A representative size distribution is shown in Fig 2A. The molality of each fragment was measured with an Agilent Bioanalyzer as previously reported [7]. The median molarity of long fragment cfDNA in patients with metastatic cancer was almost twice as high as in patients without metastases (20.0 pmol/l vs. 11.1 pmol/l), Hodges-Lehmann estimate 8.3 (95%CI 0.8–15.9; p = 0.031; Fig 2B), whereas the difference in molarity of short fragment cfDNA between patients with and without metastasis was smaller

**Table 1. Participant characteristics.**

| Group | | | n = 130 |
|---|---|---|---|
| **Patients with lung cancer** | | | **92** |
| | Adenocarcinoma | | 60 |
| | Squamous cell carcinoma | | 20 |
| | Non-small cell carcinoma not otherwise specified | | 8 |
| | Small cell carcinoma | | 4 |
| | Stage | I | 42 |
| | | II | 11 |
| | | III | 16 |
| | | IV | 22 |
| | | Unknown | 1 |
| | Without extrathoracic metastasis | | 73 |
| | With extrathoracic metastasis | | 18 |
| | Metastatic status unknown | | 1 |
| | *EGFR* mutation status (adenocarcinoma) | | |
| | | Ex19 deletion | 8 |
| | | L858R | 14 |
| | | Negative or unknown | 38 |
| **Patients with benign pulmonary disease**\* | | | **18** |
| **Healthy individuals** | | | **20** |

\* Bacterial pneumonia: 3, Lung abscess: 1, Nontuberculous mycobacterial disease: 3, Tuberculosis: 2, Aspergillosis: 2, Cryptogenic organizing pneumonia: 1, Chronic eosinophilic pneumonia: 1, Sarcoidosis: 1, Pneumothorax: 1, Others: 3

and not statistically significant (880.5 pmol/l vs. 523.7 pmol/l; Fig 2C). These results suggest the possibility of an association between the presence of long fragment cfDNA and extrathoracic metastasis.

## Long fragment DNA detected in same fraction as EVs in conditioned media of lung cancer cell lines

Next, we hypothesized that EVs might play a role in protecting long fragment cfDNA from enzyme-mediated degradation in the blood. Previous studies have reported that the size distribution of DNA from EVs derived from human plasma is located around 6 Kb [14]. EVs have been classified into small EVs (exosomes) and large EVs (oncosomes) according to their sizes, and they have been separated by centrifugation [15–17]. In addition, oncosomes have been reported to contain more tumor-derived long fragment DNA than exosomes [10]. By the protocol shown in Fig 3A, we were able to observe EVs with a transmission electron microscope (TEM) in each of the 10K and 100K pellets from conditioned media of lung cancer cell lines. Although 10K pellets contained small and large EVs, 100K pellets contained only small EVs (Fig 3B). Western blotting analysis of 10K and 100K pellets in lung cancer cell lines was based on CD9 and CD63 as markers of small EVs (exosomes), and BiP and Annexin A1 as markers of large EVs (oncosomes) [10,18]. The 10K pellets tended to contain fewer exosomes and more large EVs than 100K pellets (Fig 3C). The DNA size distribution of 10K and 100K pellets in lung cancer cell lines comprised only long fragments (Fig 3D). These results show that both small and large EVs are associated with long fragment DNA in lung cancer cell lines.

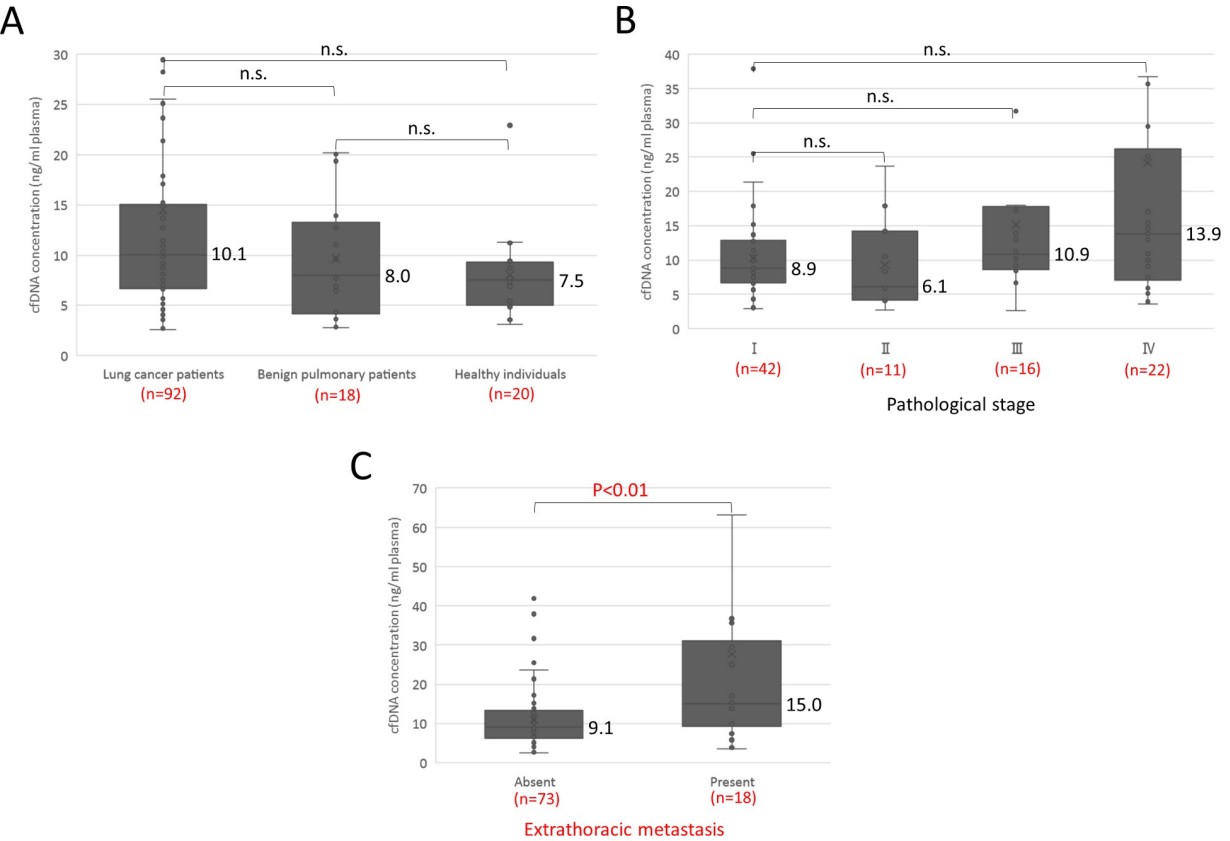

**Fig 1. cfDNA concentration in patients with lung cancer or benign pulmonary disease and in healthy individuals.** (A) Comparison of cfDNA concentration among the three participant groups. (B) Comparison of cfDNA concentration by pathological stage of lung cancer. (C) Comparison of cfDNA concentration in patients with lung cancer according to presence or absence of distant metastasis.

## Short and long fragment cfDNA also detected in plasma of mice with metastases

Next, we investigated whether long fragment DNA was detected in mice with metastases. We previously established an animal model reflecting human metastatic lung cancers by using NOD/SCID/Jak3$^{null}$ (NOJ) mice, which exhibit deficiencies in NK cell activity, macrophage and dendritic cell function, and complement activation, as well as T and B cell deficiencies [12]. H1975 cells ($1 \times 10^7$) were injected into the dorsal flanks of 24 NOJ mice. Six mice were sacrificed monthly from one to four months after injection to evaluate metastatic status and size distribution of cfDNA. Among 24 mice, quality of cfDNA was good enough for analysis of size distribution in 20 mice, in which 12 of the mice had metastases and 8 did not. Therefore, we compared size distribution of cfDNA between presence and absence of metastasis. The size distribution of cfDNA from animals without and with metastases is shown in Fig 4A and 4B. Although the amount of short fragment cfDNA was negligible in plasma of mice without metastases, as measured with capillary electrophoresis (Fig 4A), both short and long fragment cfDNA were clearly present in plasma of mice with metastases (Fig 4B). Examination of quantity showed that the median molarity of long fragment cfDNA in animals with metastases was more than twice as high than in animals without metastases (29.9 pmol/l vs. 12.2 pmol/l), Hodges-Lehmann estimate 16.6 (95%CI 2.8–54.6) p = 0.016 (Fig 4C). Median molarity of

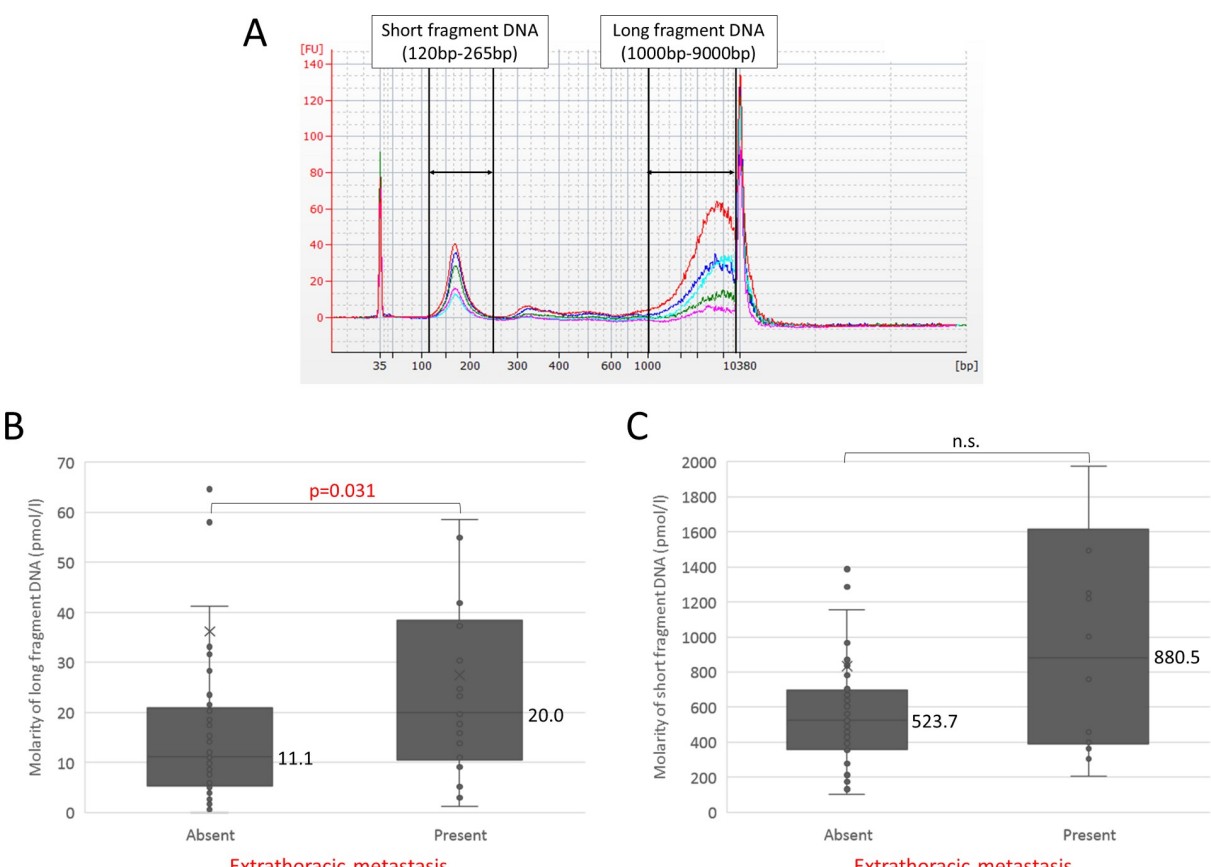

**Fig 2. Size distribution of cfDNA analyzed with Agilent Bioanalyzer among patients with lung cancer.** (A) Size distribution among 5 selected patients. Size distribution of cfDNA fragments in patients with lung cancer is bimodal with peaks around 170 bp (short) and 5 Kb (long). We defined "short fragment cfDNA" as DNA of size 120~265 bp and "long fragment cfDNA" as 1000~9000 bp. The concentration and molarity of each specified region were normalized by lower (35 bp) and upper (10380 bp) marker buffers. (B) Comparison of molarity of long fragment cfDNA according to presence or absence of distant metastasis. (C) Comparison of molarity of short fragment cfDNA according to presence or absence of distant metastasis.

short fragment cfDNA in animals with metastases was substantially higher than in animals without metastases (1185.0 pmol/l vs. 103.8 pmol/l), Hodges-Lehmann estimate 819.6 (95%CI 78.5−1184.5) p = 0.020 (Fig 4D). These results show that cancer cell-derived DNA could be degradated in peripheral blood, since ctDNA was contained in both fragments, which has been previously reported [7].

## Allele frequency of L858R mutation higher in short fragment cfDNA than in long fragment cfDNA

To investigate the origins of long and short fragments of cfDNA in human, we separated plasma of lung cancer patients with L858R mutation by the same centrifugation protocol as conditioned media of lung cancer cell lines (Fig 5A), and we found long fragment cfDNA was mainly identified in 10K pellet and short fragment was detected in supernatant (Fig 5B). To compare the amount of short and long fragment tumor-derived DNA in patients with lung cancer, cfDNA in plasma collected from 13 patients whose cancer had L858R mutations was separated into short and long fragments by centrifugation. First, we compared the cfDNA concentration, molarity of short fragment cfDNA, molarity of long fragment cfDNA, and molarity

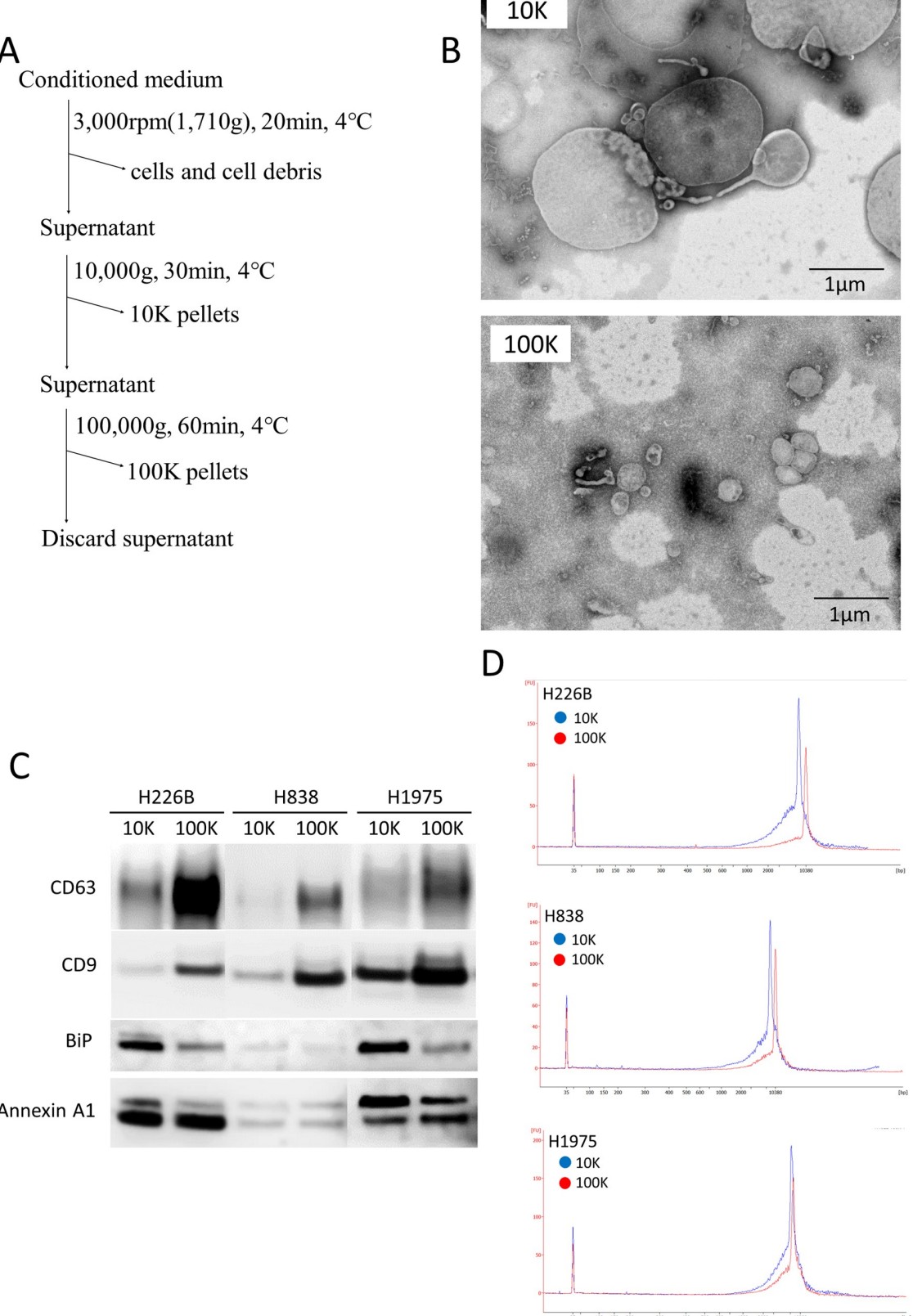

**Fig 3. Association of long fragment DNA with extracellular vesicles (EVs) in lung cancer cell lines.** (A) Protocol for EV isolation using conditioned media of lung cancer cell lines. (B) Images of 10K and 100K pellets with a transmission electron microscope (TEM). (C) Western blotting analysis of 10K and 100K pellets. 5 μg of EV lysate was applied. We used antibodies to CD9 and CD63 as markers of small EVs (exosomes), and BiP and Annexin A1 as markers of large EVs. (D) DNA size distribution among 10K and 100K pellets.

of combined short and long fragment cfDNA among the four sample types: plasma, 10K pellets, 100K pellets, and supernatant (S1 Fig). Second, we compared the correlation of L858R copy number detected by droplet digital PCR (ddPCR) with the four measures: cfDNA concentration, molarity of short fragment cfDNA, molarity of long fragment cfDNA, and molarity of combined short and long fragment cfDNA (Fig 5C–5F). L858R copy number was more strongly correlated with cfDNA concentration, molarity of short fragment cfDNA, and molarity of combined short and long fragment cfDNA than with molarity of long fragment cfDNA alone. In addition, the allele frequency of L858R mutation in cfDNA of supernatant tended to be the highest among all sample types (Fig 5G). The allele frequency of L858R mutation was pronounced in cfDNA of supernatant but effectively zero in 10K pellets (13.0% vs 0.0%), Hodges-Lehmann estimate 12.3 (95%CI 1.1–21.3) p<0.01 (Fig 5G). Therefore, tumor-derived DNA from patients with lung cancer is present in significantly higher amounts in short fragment cfDNA than in long fragment cfDNA.

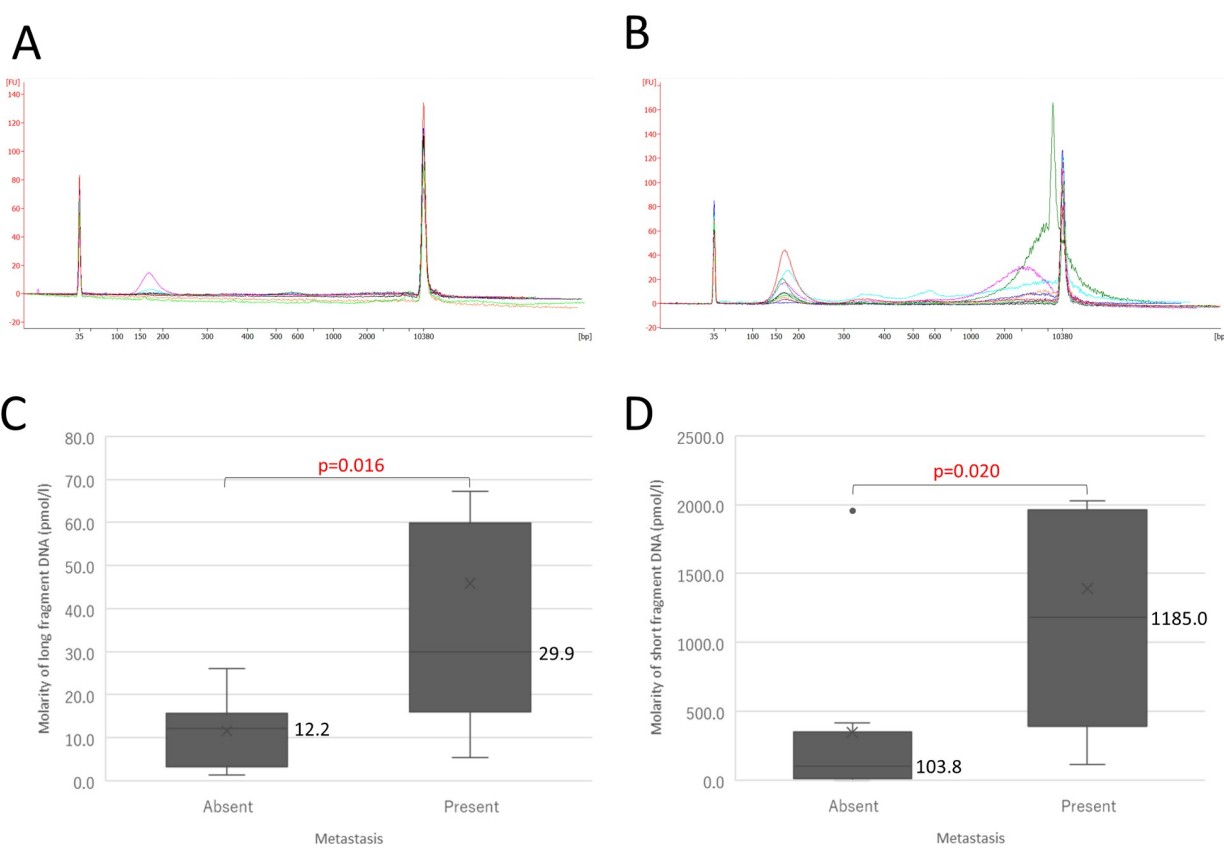

**Fig 4. Size distribution of cfDNA in mice injected with H1975 cells.** (A) Size distribution of cfDNA in mice without metastases. (B) Size distribution of cfDNA in mice with metastases. (C) Comparison of molarity of long fragment cfDNA in mice according to presence or absence of distant metastases. (D) Comparison of molarity of short fragment cfDNA in mice according to presence or absence of distant metastasis.

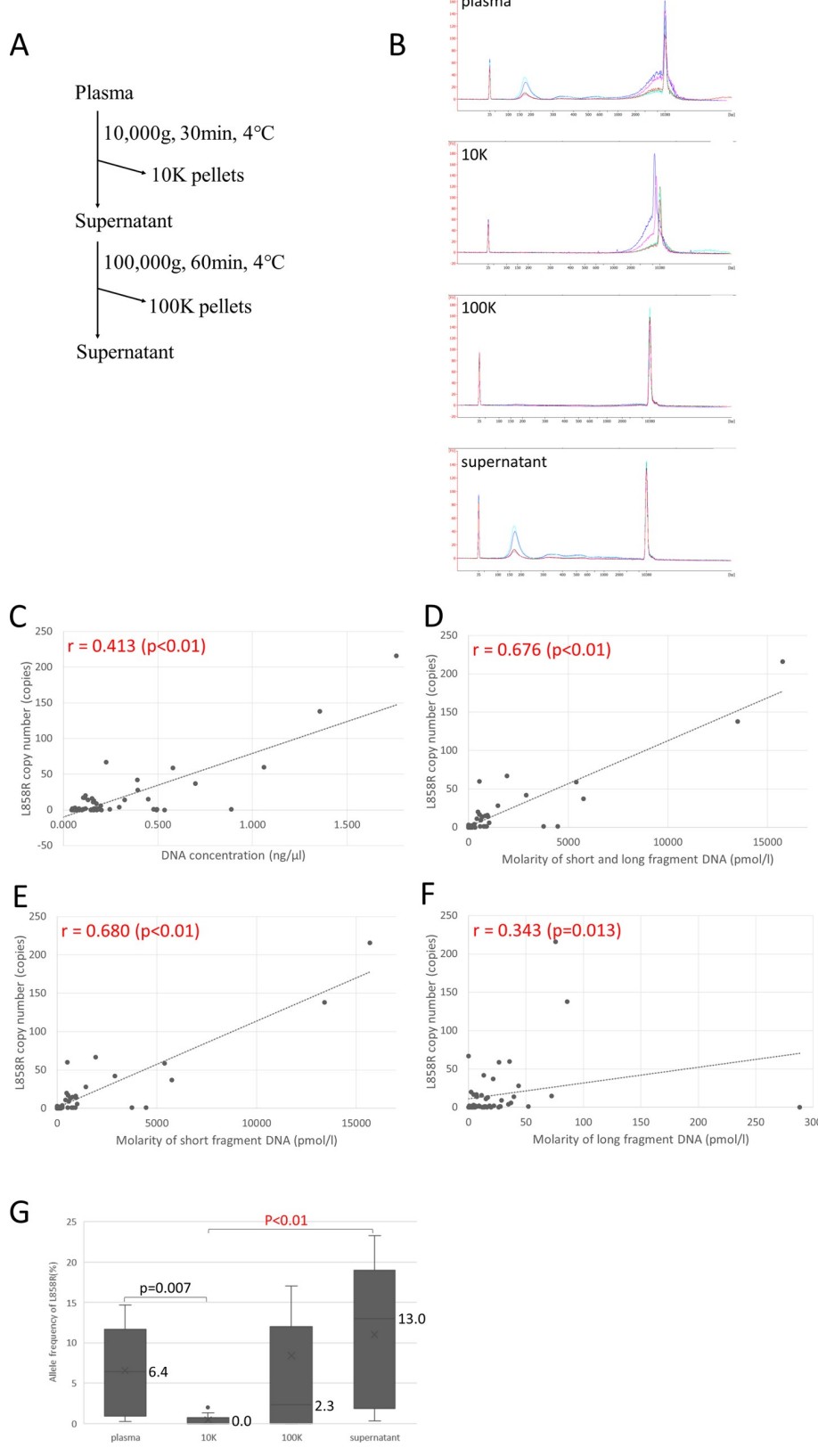

**Fig 5. Allele frequency of *EGFR* L858R mutation of short and long fragment cfDNA in patients with lung cancer.**
(A) The protocol to isolate short and long fragments of cfDNA in plasma from patients with lung cancer patients. (B)
A representative result of the size distribution obtained with Bioanalyzer. The results of plasma, 10K pellet, 100K pellet,
and supernatant are shown. (C-F) Correlation of L858R copy number with the four measures: cfDNA concentration,
molarity of short fragment cfDNA, molarity of long fragment cfDNA, and molarity of combined short and long
fragment cfDNA. (G) Comparison among sample types of the allele frequency of *EGFR* L858R mutation by using
droplet digital PCR (ddPCR).

## Discussion

In this paper, we have shown that, in patients with lung cancer, total amount of cfDNA was
higher in those who had distant metastasis than in those who did not. In particular, the
amount of long fragment cfDNA was significantly higher in those patients. Long fragment
cfDNA was present together with EVs in the fraction separated by centrifugation of cell culture
media. In human plasma samples, long fragment cfDNA was observed in the same fraction as
conditioned media, whereas short fragment cfDNA existed in the supernatant after centrifuga-
tion at 100,000 g, in which the amount of ctDNA was two times higher than in plasma isolated
by a conventional procedure.

The total amount of cfDNA is known to be correlated with tumor burden, stage, number of
metastatic sites, and worse overall survival [19–21]. Most papers describe cfDNA at size 160–
170 bp, and long fragment cfDNA has been thought to be contamination by genomic DNA.
We previously reported a pre-analytical procedure for assessing cfDNA, in which we con-
firmed that long fragment cfDNA appeared in samples stored under inappropriate conditions
[8]. However, long fragment cfDNA of 5 Kb was definitely observed in patients with advanced
lung cancer even under proper conditions, as described in the present paper and our previous
paper [7]. In cancer cell culture media, only long fragment cfDNA was observed, but in an ani-
mal model using the same cancer cells we found both long and short fragment cfDNA. These
results suggest that cancer cell-derived DNA could be degraded in peripheral blood, since
ctDNA was contained in both fragments.

Recently, large EVs, named oncosomes, have been reported to contain large size DNA [10].
EVs vary in size, and the mechanism of their release also differs between small EVs (exosomes)
and large EVs (oncosomes). The former are released through fusion of the multi-vesicular
body and the plasma membrane, whereas the latter are released directly from the plasma mem-
brane by blebbing and budding mechanisms [22] and contain extra-nuclear large size DNA.
Long fragment cfDNA has been reported to exist in oncosomes isolated from patients with
metastatic prostate cancer, but not in oncosomes isolated from cancer-free individuals [10].
Our results indicate that long fragment cfDNA is detected in the same fraction as EVs—
including oncosomes and exsosomes—after centrifugation, and short fragment cfDNA was
observed after 100,000 g centrifugation, suggesting that association with EVs helps it avoid
degradation. The long fragment cfDNA contained little ctDNA on the basis of on the basis of
*EGFR* driver mutation L858R as an indicator of ctDNA. EVs mainly contained miRNA and
proteins, which are known to be metastasis-associated factors such as brain-derived neuro-
trophic factor (BDNF), C-X-C motif chemokine 12 (CXCL12), and osteopontin, and are
potent mediators of communication between tumor cells and stroma [23]. These mediators
may contribute to tumor progression, and long fragment cfDNA exists along with these fac-
tors. Whether long fragments have a role in tumor progression should be further investigated.

The possible origin of cfDNA is schematically summarized in Fig 6. cfDNA, including
ctDNA, is released from cancer cells, and is mainly degraded into nucleosome units in periph-
eral blood. cfDNA co-existing with EVs escapes degradation and remains as long fragment
cfDNA, although it is not clarified whether cfDNA is inside or outside of the EVs. We also

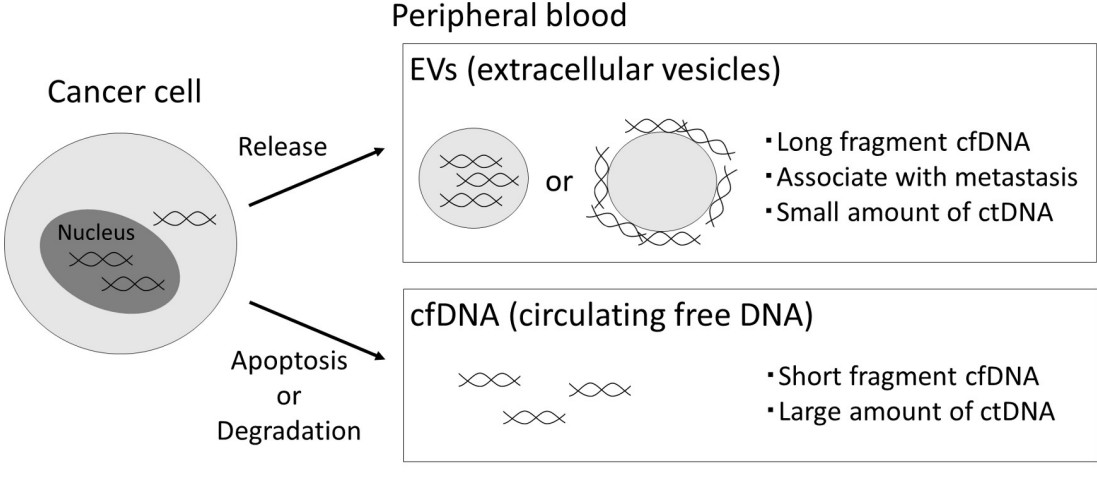

**Fig 6. The possible origin of cfDNA in cancer patients.** cfDNA, including ctDNA, is released from cancer cells, and is mainly degraded in peripheral blood. cfDNA co-existing with EVs escapes degradation and remains as long fragment cfDNA, although it is not clarified whether cfDNA is inside or outside of the EVs.

found that the amount of ctDNA is greater in the fraction after centrifugation at 100,000 g than after the conventional method. Recently, median length cfDNA fragments from patients with cancer has been reported to be around 163 bp by analysis of its fragmentation pattern using whole genome sequencing [24]. We could not analyze the precise size of ctDNA contained in cfDNA better than with the conventional procedure, but it could be of smaller size since the fraction was obtained with higher-speed centrifugation.

Although we hypothesized that ctDNA is contained in long fragment cfDNA, our results showed that it predominantly exists in short fragment cfDNA. However, we also found that performing just two rounds of centrifugation apparently leads to more effective isolation of ctDNA. In the next step, we need to confirm the usefulness of the method by using a larger number of clinical samples.

## Supporting information

**S1 Fig. Comparison of DNA concentration and molarity.** (A-D) Comparison of the DNA concentration, molarity of short fragment DNA, molarity of long fragment DNA, and molarity of combined short and long fragment DNA among plasma, 10K pellets, 100K pellets, and supernatant.
(TIF)

**S1 Table. The files of the statistical analyses.**
(DOCX)

**S1 Raw image.**
(TIF)

**S2 Raw image.**
(TIF)

## Author Contributions

**Conceptualization:** Tomonori Abe, Chiho Nakashima, Akemi Sato, Naoko Sueoka-Aragane.

**Data curation:** Tomonori Abe, Chiho Nakashima, Akemi Sato, Yohei Harada, Naoko Sueoka-Aragane.

**Formal analysis:** Atsushi Kawaguchi.

**Investigation:** Tomonori Abe, Chiho Nakashima, Akemi Sato, Yohei Harada.

**Methodology:** Tomonori Abe, Chiho Nakashima, Akemi Sato, Naoko Sueoka-Aragane.

**Supervision:** Shinya Kimura, Naoko Sueoka-Aragane.

**Validation:** Tomonori Abe, Chiho Nakashima, Akemi Sato, Yohei Harada.

**Writing – original draft:** Tomonori Abe, Chiho Nakashima, Naoko Sueoka-Aragane.

**Writing – review & editing:** Tomonori Abe, Chiho Nakashima, Akemi Sato, Yohei Harada, Eisaburo Sueoka, Shinya Kimura, Naoko Sueoka-Aragane.

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
