## [Decision Letter · Decision Letter 0]

30 Mar 2020

PONE-D-19-35699

Origin of circulating free DNA in patients with lung cancer

PLOS ONE

Dear Dr. Abe,

Thank you for submitting your manuscript to PLOS ONE. After careful consideration, we feel that it has merit but does not fully meet PLOS ONE’s publication criteria as it currently stands. Therefore, we invite you to submit a revised version of the manuscript that addresses the points raised during the review process.

Please address the reviewers significant concerns. Given that one is a reject and the other major revision, I will be looking for a very good response. 

We would appreciate receiving your revised manuscript by May 14 2020 11:59PM. To enhance the reproducibility of your results, we recommend that if applicable you deposit your laboratory protocols in protocols.io, where a protocol can be assigned its own identifier (DOI) such that it can be cited independently in the future. For instructions see: http://journals.plos.org/plosone/s/submission-guidelines#loc-laboratory-protocols

We look forward to receiving your revised manuscript.

Kind regards,

Jeffrey Chalmers, Ph.D.

Academic Editor

PLOS ONE

Journal Requirements:

2. To comply with PLOS ONE submission requirements, in your Methods section, please provide additional information regarding the experiments involving animals and ensure you have included details on (1) methods of sacrifice, (2) methods of anesthesia and/or analgesia, and (3) efforts to alleviate suffering.

3. In your Methods section, please provide additional information about the participant recruitment method and the demographic details of your participants. Please ensure you have provided sufficient details to replicate the analyses such as a description of any inclusion/exclusion criteria that were applied to participant recruitment and a description of how participants and healthy volunteers were recruited.

Reviewers' comments:

Reviewer's Responses to Questions

**Comments to the Author**

1. Is the manuscript technically sound, and do the data support the conclusions?

Reviewer #1: Yes

Reviewer #2: Partly

2. Has the statistical analysis been performed appropriately and rigorously? 

Reviewer #1: Yes

Reviewer #2: No

3. Have the authors made all data underlying the findings in their manuscript fully available?

Reviewer #1: Yes

Reviewer #2: Yes

4. Is the manuscript presented in an intelligible fashion and written in standard English?

Reviewer #1: Yes

Reviewer #2: Yes

5. Review Comments to the Author

Reviewer #1: This is a translational study on the origin of cfDNA in lung cancer. Authors utilized both cell-line/animal model and plasma samples from human subjects. The experiments were conducted in high quality and manuscript was written clearly. Authors have reported the long fragment cfDNA to be associated with distant metastasis but not the short fragment cfDNA. This is interesting but I do have a number of queries as followed:

1. Table 1: 22 patients had stage IV disease while only 18 had distant metastasis. Thus 4 patients with stage IV disease actually didn’t have M1 disease. Please note that pleural effusion is considered distant metastasis. Please clarify.

2. Results, first paragraph (line 265): Subheading is misleading. There is no statistical difference in total cfDNA between patient with lung cancer and benign pulmonary disease/healthy individual. Author must change the subheading.

3. Figure 1: please provide sample size for each group.

4. There was difference in both short fragment cfDNA (880 vs 523 pmol/l) and long fragment cfDNA (20 vs 11 pmol/l) between patients with and without metastasis. Only statistically significance was reported to be negative for the short fragment cfDNA subgroup. We will need statistical review on this and certainly authors’ statement of “long fragment cfDNA was more clearly associated with distant metastasis than short fragment cfDNA” is NOT strongly supported by the data.

5. Total of 20 mice were analyzed and 12 with metastasis and 8 without. How did the authors evaluate for metastasis? Did they perform CT scan on the mice or did they just examine all the organs after sacrificing the mice? If for the latter, did they include examination of the CNS?

6. The statement “Although only short fragment cfDNA was detected in plasma of mice without metastases (Fig 4A), both short and long fragment cfDNA was detected in plasma of mice with metastases (Fig 4B).” is a bit confusing. As per Figure 4C and 4D, both short and long fragment cfDNA were present in mice with and without distant metastasis. Only the quantities were different. Please clarify

7. Please provide the EGFR mutation status of lung cancer patients in Table 1.

8. Why would the authors analyse only the EGFR 21 L858R mutation? What is the total sample size? Why ignore the testing of EGFR exon 19 deletion (which is also easily analysed by ddPCR)?

9. The statement, “Since long fragment was detected in same fraction as that observed in cancer cell lines, long fragment DNA was suggested to be associated with EVs” is speculative. cfDNA from cell line would not be metabolized and excreted by kidney, while cfDNA in human subjects are. In absence of data on metabolization and excretion of long vs short fragment DNA, we cannot make direct analogy with observation in cell lines.

Reviewer #2: This is an interesting paper about the origin of circulating tumor DNA (ctDNA) in patients with lung cancer.

major Critiques:

1. The authors should clearly state the primary objective, central hypothesis, and primary outcome variable of this research.

2. Please shorten this manuscript.

3. In addition to p-values, the authors should report the 95% confidence intervals (CIs) for all major findings.

4. The authors defined “short fragment cfDNA” as 126 DNA of size 120~265 bp and “long fragment cfDNA” as 1000~9000 bp without any justification. Please justify it.

5. The discussion of the normalization method was superficial. Please provide a detailed justification of it.

6. The authors stated that “multiple pairwise comparisons were made”; however, they did not clearly specify the method that they used. In addition, it is unclear which p-values reported in this paper were adjusted for the multiple comparisons.

7. The authors should report the non-parametric Spearman correlation coefficient instead of Pearson correlation coefficient.

8. The log files of the statistical analyses (not the results) should be added to the supplemental data. This will help readers understand the detailed statistical analysis procedures.

6. PLOS authors have the option to publish the peer review history of their article (what does this mean?). If published, this will include your full peer review and any attached files.

Reviewer #1: No

Reviewer #2: No

---

## [Author Response · Author response to Decision Letter 0]

29 May 2020

Dear Dr. Chalmers:

I am submitting a revised version of our manuscript entitled, "Origin of circulating free DNA in patients with lung cancer". 

We have attempted to address all of the points raised by the reviewers. We highlighted corresponding manuscript changes and corrections in red. We also found a few mistakes, which we corrected (also in red). Specific responses follow.

Responses to Reviewer #1

Q1) Table 1: 22 patients had stage IV disease while only 18 had distant metastasis. Thus 4 patients with stage IV disease actually didn’t have M1 disease. Please note that pleural effusion is considered distant metastasis. Please clarify.

A1) As the reviewer points out, the motivation for this is not obvious. We therefore added a sentence, and changed Table 1 and Figure 1C, as follows.

Line 231: Since we have reported that the frequency of ctDNA was higher in distant metastasis than in the intrathoracic region on the basis of a prospective observational study (13), we compared patients with and without extrathoracic metastases. Among patients with lung cancer the median concentration of total cfDNA in those with extrathoracic metastases was significantly…..

Table1

Group n=130

Patients with lung cancer 92

 Adenocarcinoma 60

 Squamous cell carcinoma 20

 Non-small cell carcinoma not otherwise specified 8

 Small cell carcinoma 4

 Stage Ⅰ 42

 Ⅱ 11

 Ⅲ 16

 Ⅳ 22

 Unknown 1

 Without extrathoracic metastasis 73

 With extrathoracic metastasis 18

 Metastatic status unknown 1

 EGFR mutation status (adenocarcinoma) 

 Ex19 deletion 8

 L858R 14

 Negative or unknown 38

Patients with benign pulmonary disease* 18

Healthy individuals 20

* Bacterial pneumonia: 3, Lung abscess: 1, Nontuberculous mycobacterial disease: 3, Tuberculosis: 2, Aspergillosis: 2, Cryptogenic organizing pneumonia: 1, Chronic eosinophilic pneumonia: 1, Sarcoidosis: 1, Pneumothorax: 1, Others: 3

Figure 1C

Q2) Results, first paragraph (line 224): Subheading is misleading. There is no statistical difference in total cfDNA between patient with lung cancer and benign pulmonary disease/healthy individual. Author must change the subheading.

A2) We changed the subheading.

Line 224: Total cfDNA higher in lung cancer with extrathoracic metastases than without

Q3) Figure 1: please provide sample size for each group.

A3) We added the sample size for each group in Figure 1 as below.

Q4) There was difference in both short fragment cfDNA (880 vs 523 pmol/l) and long fragment cfDNA (20 vs 11 pmol/l) between patients with and without metastasis. Only statistically significance was reported to be negative for the short fragment cfDNA subgroup. We will need statistical review on this and certainly authors’ statement of “long fragment cfDNA was more clearly associated with distant metastasis than short fragment cfDNA” is NOT strongly supported by the data.

A4) We changed the sentence, and we corrected Figures 2B and C as shown below.

Line 247: These results suggest the possibility of an association between the presence of long fragment cfDNA and extrathoracic metastasis.

Figures 2B and C

Q5) Total of 20 mice were analyzed and 12 with metastasis and 8 without. How did the authors evaluate for metastasis? Did they perform CT scan on the mice or did they just examine all the organs after sacrificing the mice? If for the latter, did they include examination of the CNS?

A5) We did not perform CT scans to evaluate metastasis. Instead, we sacrificed and dissected the mice when either body weight decreased by 10% of original weight or metastatic manifestations (such as superficial lymph node swelling or ascites) were observed, as mentioned in Line 181. We conducted systematic examination for metastases, including in the brain. 

Q6) The statement “Although only short fragment cfDNA was detected in plasma of mice without metastases (Fig 4A), both short and long fragment cfDNA was detected in plasma of mice with metastases (Fig 4B).” is a bit confusing. As per Figure 4C and 4D, both short and long fragment cfDNA were present in mice with and without distant metastasis. Only the quantities were different. Please clarify

A6) We changed the sentences as follows to avoid causing confusion.

Line 306: Although the amount of short fragment cfDNA was negligible in plasma of mice without extrathoracic metastases, as measured with capillary electrophoresis (Fig 4A), both short and long fragment cfDNA were clearly present in plasma of mice with metastases (Fig 4B). Examination of quantity showed that ……

Q7) Please provide the EGFR mutation status of lung cancer patients in Table 1.

A7) We added EGFR mutation status in Table 1, as shown above (answer to Q1).

Q8) Why would the authors analyse only the EGFR 21 L858R mutation? What is the total sample size? Why ignore the testing of EGFR exon 19 deletion (which is also easily analysed by ddPCR)?

A8) To investigate the content of ctDNA in each faction, at least 2 ml plasma was required. Among 22 plasma samples with over 2ml remaining, 17 were L858R positive and 5 had exon 19 deletions in tissue specimens. If the amount of ctDNA is small, it is sometimes not detected because of degradation; ddPCR could successfully be performed in 14 of the 22 blood samples. The results comprised 13 L858R mutations and 1 exon 19 deletion. Therefore, we analyzed the L858R positive samples for comparison of quantity in each fraction after centrifugation. 

Q9) The statement, “Since long fragment was detected in same fraction as that observed in cancer cell lines, long fragment DNA was suggested to be associated with EVs” is speculative. cfDNA from cell line would not be metabolized and excreted by kidney, while cfDNA in human subjects are. In absence of data on metabolization and excretion of long vs short fragment DNA, we cannot make direct analogy with observation in cell lines.

A9) Ideally, we should show co-existence of long fragment cfDNA and EV in human plasma samples. However, detection of EV in human samples by TEM was difficult because the amount of EV was too small. Therefore, we used conditioned medium of cancer cell lines to show EV in the long fragment fraction by TEM. 

As the reviewer suggests, association between long fragment and EV is speculative. Therefore, we deleted from line 330 the sentence, “Since long fragment was detected in same fraction as that observed in cancer cell lines, long fragment DNA was suggested to be associated with EVs.”. Also, we changed the title of Figure 6 to “ Possible origin of …”, and added the following sentence.

Line 404: The possible origin of cfDNA is schematically summarized in Fig 6.

Responses to Reviewer #2

Q1) The authors should clearly state the primary objective, central hypothesis, and primary outcome variable of this research.

A1)　We added the following sentences at the ends of the Introduction and Discussion sections to clarify the primary objective, hypothesis, and principal outcome variable of this research.

Line 73: The rapid progress of technology for liquid biopsy has led to high sensitivity of ctDNA detection, but it is still limited to advanced cancers. It is now necessary to improve the effectiveness of the ctDNA isolation system to allow detection even among patients who are in the early stages of cancer. Based on our previous results, we hypothesized that long fragment cfDNA observed in patients with advanced lung cancer might be contained in oncosomes, so that purification of ctDNA derived from oncosomes could lead to improved sensitivity for detecting ctDNA. 

Line 415: Although we hypothesized that ctDNA is contained in long fragment cfDNA, our results showed that it predominantly exists in short fragment cfDNA. However, we also found that performing just two rounds of centrifugation apparently leads to more effective isolation of ctDNA. In the next step, we need to confirm the usefulness of the method by using a larger number of clinical samples.

Q2) Please shorten this manuscript.

A2)　We deleted some sentences in the Introduction and Discussion sections to remove redundant parts.

Q3) In addition to p-values, the authors should report the 95% confidence intervals (CIs) for all major findings.

A3)　We used JMP® analysis (SAS Institute Inc., NC, US) to compute 95% confidence intervals (CIs) with the cooperation of Prof. Kawaguchi, a biostatistician in Saga University. The Hodges-Lehmann method was used to estimate median differences and 95% CIs for comparison of cfDNA between absence and presence of extrathoracic metastasis overall (Fig. 1C) and for long or short fragments (Fig. 2B, C, Fig. 4C, D). The Steel-Dwass test was used for analysis of multiple pairwise comparisons of allele frequency of L858R among each fraction (Fig. 5G).

We added the following sentences and added Prof. Kawaguchi as a co-author.

Line 216: If the outcome was statistically significant, multiple pairwise comparisons were made with the Steel-Dwass test. The Hodges-Lehmann method was used to estimate median differences and 95% confidence intervals (CIs) for comparing the two groups.

Line 236: Hodges-Lehmann estimate 6.1 (95%CI 1.5�11.1) p<0.01 (Fig 1C) 

Line 244: Hodges-Lehmann estimate 8.3 (95%CI 0.8�15.9) p=0.031 (Fig 2B),

Line 311: Hodges-Lehmann estimate 16.6 (95%CI 2.8�54.6) p=0.016 (Fig 4C).

Line 314: Hodges-Lehmann estimate 819.6 (95%CI 78.5�1184.5) p=0.020 (Fig 4D).

Line 348: Hodges-Lehmann estimate (95%CI 1.1�21.3) p<0.01 (Fig 5G).

Fig 4C, D

Fig 5G

Q4) The authors defined “short fragment cfDNA” as DNA of size 120~265 bp and “long fragment cfDNA” as 1000~9000 bp without any justification. Please justify it.

A4) Our previous analysis showed that two peaks of cfDNA were consistently observed, in the 120~265 bp and 1000~9000 bp ranges (reference No.7). Therefore, we defined “short fragment cfDNA” as DNA of size 120~265 bp and “long fragment cfDNA” as 1000~9000 bp.

We changed the following sentence.

Line 130: …“long fragment cfDNA” as 1000~9000 bp, as reported previously (7). 

Q5) The discussion of the normalization method was superficial. Please provide a detailed justification of it.

A5) According to Agilent’s instruction, we changed to the following sentences.

Line 131: The concentration and molarity of each specified region were automatically calculated against internal standards by Agilent 2100 Expert software version B.02.08.

Q6) The authors stated that “multiple pairwise comparisons were made”; however, they did not clearly specify the method that they used. In addition, it is unclear which p-values reported in this paper were adjusted for the multiple comparisons.

A6) If the outcome was statistically significant, multiple pairwise comparisons were made with the Steel-Dwass test to compute 95% CI as mentioned in A3, and we added the following sentence.

Line 216: If the outcome was statistically significant, multiple pairwise comparisons were made with the Steel-Dwass test.

Q7) The authors should report the non-parametric Spearman correlation coefficient instead of Pearson correlation coefficient.

A7) We changed the method of calculating the correlation coefficient, as suggested. We changed Figure 5 C,D,E,F, and added the following sentence.

Line 220: groups was assessed with the non-parametric Spearman correlation coefficient.

Q8) The log files of the statistical analyses (not the results) should be added to the supplemental data. This will help readers understand the detailed statistical analysis procedures.

A8) We added the files of the statistical analyses as a supplemental Table.

Sincerely yours,

Naoko Sueoka-Aragane, MD, PhD

Division of Hematology, Respiratory Medicine and Oncology

Department of Internal Medicine

Faculty of Medicine, Saga University

---

## [Decision Letter · Decision Letter 1]

19 Jun 2020

Origin of circulating free DNA in patients with lung cancer

PONE-D-19-35699R1

Dear Dr. Abe,

We’re pleased to inform you that your manuscript has been judged scientifically suitable for publication and will be formally accepted for publication once it meets all outstanding technical requirements.

Kind regards,

Jeffrey Chalmers, Ph.D.

Academic Editor

PLOS ONE

Additional Editor Comments (optional):

Reviewers' comments:

Reviewer's Responses to Questions

**Comments to the Author**

1. If the authors have adequately addressed your comments raised in a previous round of review and you feel that this manuscript is now acceptable for publication, you may indicate that here to bypass the “Comments to the Author” section, enter your conflict of interest statement in the “Confidential to Editor” section, and submit your "Accept" recommendation.

Reviewer #2: All comments have been addressed

2. Is the manuscript technically sound, and do the data support the conclusions?

Reviewer #2: Yes

3. Has the statistical analysis been performed appropriately and rigorously? 

Reviewer #2: Yes

4. Have the authors made all data underlying the findings in their manuscript fully available?

Reviewer #2: Yes

5. Is the manuscript presented in an intelligible fashion and written in standard English?

Reviewer #2: Yes

6. Review Comments to the Author

Reviewer #2: The authors have responded well to the statistical issues raised in the previous review. There is no further statistical concern about this revised manuscript.

7. PLOS authors have the option to publish the peer review history of their article (what does this mean?). If published, this will include your full peer review and any attached files.

Reviewer #2: No

---

## [Editor Report · Acceptance letter]

25 Jun 2020

PONE-D-19-35699R1 

Origin of circulating free DNA in patients with lung cancer 

Dear Dr. Abe:

I'm pleased to inform you that your manuscript has been deemed suitable for publication in PLOS ONE. Congratulations! Your manuscript is now with our production department. 

Kind regards, 

on behalf of

Dr. Jeffrey Chalmers 

Academic Editor

PLOS ONE